# Periodically Released Magmatic Fluids Create a Texture of Unidirectional Solidification (UST) in Ore-Forming Granite: A Fluid and Melt Inclusion Study of W-Mo Forming Sannae-Eonyang Granite, Korea

**Jung Hun Seo** [1,*] , **Yevgeniya Kim** [1], **Tongha Lee** [1] **and Marcel Guillong** [2]

1    Department of Energy and Resources Engineering, Inha University, Inharo 100, Michuhol-Gu,
     Incheon 22212, Korea; evgeshik.kim@gmail.com (Y.K.); eastriver827@gmail.com (T.L.)
2    Institute of Geochemistry and Petrology, ETH Zurich, Clausiusstrasse 25, CH-8092 Zurich, Switzerland;
     marcel.guillong@erdw.ethz.ch
*    Correspondence: seo@inha.ac.kr; Tel.: +82-328-607-557

**Abstract:** The Upper Cretaceous Sannae-Eonyang granite crystallized approximately 73 Ma and hosted the Sannae W-Mo deposit in the west and the Eonyang amethyst deposit in the east. The granite contained textural zones of miarolitic cavities and unidirectional solidification texture (UST) quartz. The UST rock sampled in the Eonyang amethyst mine consisted of (1) early cavity-bearing aplitic granite, (2) co-crystallization of feldspars and quartz in a granophyric granite, and (3) the latest unidirectional growth of larger quartz crystals with clear zonation patterns. After the UST quartz was deposited, aplite or porphyritic granite was formed, repeating the prior sequence. Fluid and melt inclusions occurring in the UST quartz and quartz phenocrysts were sampled and studied to understand the magmatic-hydrothermal processes controlling UST formation and W-Mo mineralization in the granite. The composition of melt inclusions in the quartz phenocrysts suggested that the UST was formed by fractionated late-stage granite. Some of the melt inclusions occurring in the early-stage UST quartz were associated with aqueous inclusions, indicating fluid exsolution from a granitic melt. Hypersaline brine inclusions allowed the calculation of the minimum trapping pressure of 80–2300 bars. Such a highly fluctuating fluid pressure might be potentially due to a lithostatic-hydrostatic transition of pressure-attending fluid loss during UST formation. Highly fluctuating lithostatic-hydrostatic pressures created by fluid exsolution allowed shifting of the stability field from a quartz-feldspar cotectic to a single-phase quartz. The compositions of brine fluid assemblages hosted in the quartz phenocrysts deviated from the fluids trapped in the UST quartz, especially regarding the Rb/Sr and Fe/Mn ratios and W and Mo concentrations. The study of melt and fluid inclusions in the Eonyang UST sample showed that the exsolution of magmatic fluid was highly periodic. A single pulse of magmatic fluids of variable salinities/densities might have created a single UST sequence, and a new batch of magmatic fluid exsolution would be required to create the next UST sequence.

**Keywords:** Eonyang; Sannae; UST; amethyst; fluid inclusions; W-Mo

## 1. Introduction

Magmatic fluid exsolution processes in the upper part of a crystallizing magma are critical for forming magmatic-hydrothermal ore systems, such as Sn-W granites and porphyry-style deposits [1–4]. Igneous textures that indicate fluid saturation and exsolution in the apical parts of granitic intrusions comprise miarolitic cavities, orbicular texture, and unidirectional solidification texture (UST) in the aplite-pegmatite interface [5–9]. UST is caused by anisotropic mineral crystallization directed toward the inner part of a magma chamber, which has been identified globally at intrusions generating economic hydrothermal deposits of Cu-Au [10–15], Mo [6,16,17], and Sn or W [18–21]. The genetic geological

link between the features of UST in magma and economic ore mineralization has been investigated by petrographic observation [5,6], mineral geochemistry [9,20], stable isotope geochemistry [7,16,19], and fluid inclusion studies [12,18]. However, many questions remain unanswered regarding the pressure-temperature (P-T) evolution of the magmatic-hydrothermal fluid. Although UST formation in the upper part of an intrusion involves a volatile fluid-phase coexisting within the magma [22], how the fluid phase evolves (in terms of composition or P-T parameters) during UST formation has not yet been fully established. Fluid inclusion studies of miarolitic cavities in potential mineralizing plutons may provide P-T and compositional information for the magmatic fluid phase that might be compositionally unrelated to mineralization, and some might be prior to mineralization [3,4,23]. Fluid inclusions in UST crystals are generally poorly preserved because many existing USTs are tiny bands of quartz, similar to curvy quartz veins. Fluid inclusions hosted in quartz crystals of UST have been studied by microthermometry [12,18], but not by microanalyses such as laser Raman spectroscopy and laser ablation (LA) ICP-MS.

The Sannae-Eonyang granite or granitoid [24,25], located in the southeastern part of the Korean Peninsula, hosts numerous miarolitic cavities [26], including the Eonyang amethyst and W-Mo mineralizations such as a Sannae deposit (Figure 1). The Sannae W-Mo deposit is a vein-type scheelite-wolframite-molybdenite-bearing deposit [27,28]. The Eonyang amethyst deposit [29–32] is associated with an aplitic part in the granite and contains a series of UST sequences consisting of quartz and feldspar, in which quartz crystals have developed a clear zonation along the c-axis (Figure 2). The quartz crystals in the USTs and phenocrysts of the associated aplite contain melt and fluid inclusions. Here, we can study the melt and fluid inclusions using microthermometry and LA-ICP-MS microanalysis to understand magmatic and magmatic-hydrothermal evolution associated with the formation of UST and W-Mo ores.

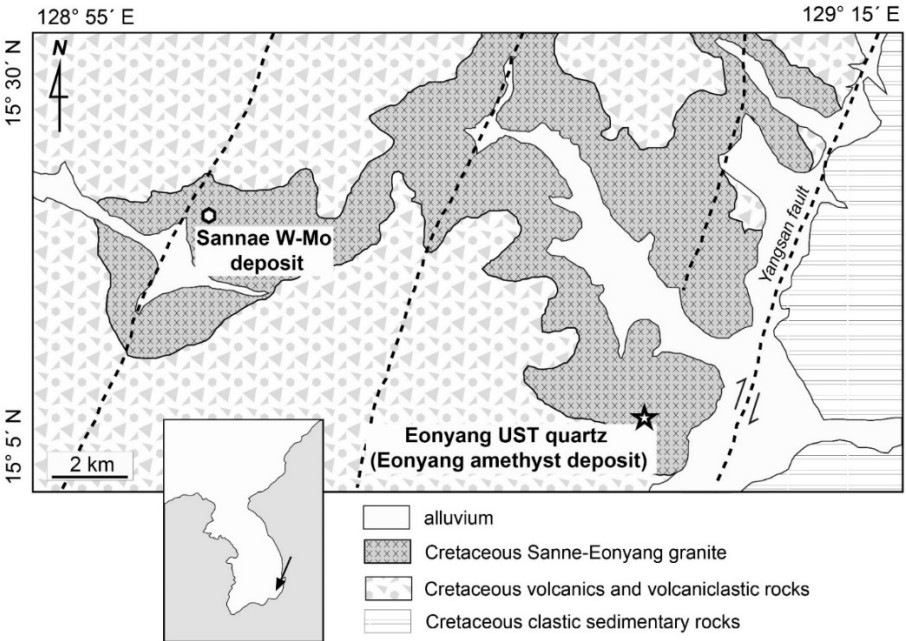

**Figure 1.** Geological map of Sannae-Eonyang granite and locations of the Eonyang amethyst and Sannae W-Mo deposit. Dashed lines represent clockwise strike-slip faults, including the Yangsan transform fault. Eonyang UST quartz is sampled from the Eonyang amethyst deposit.

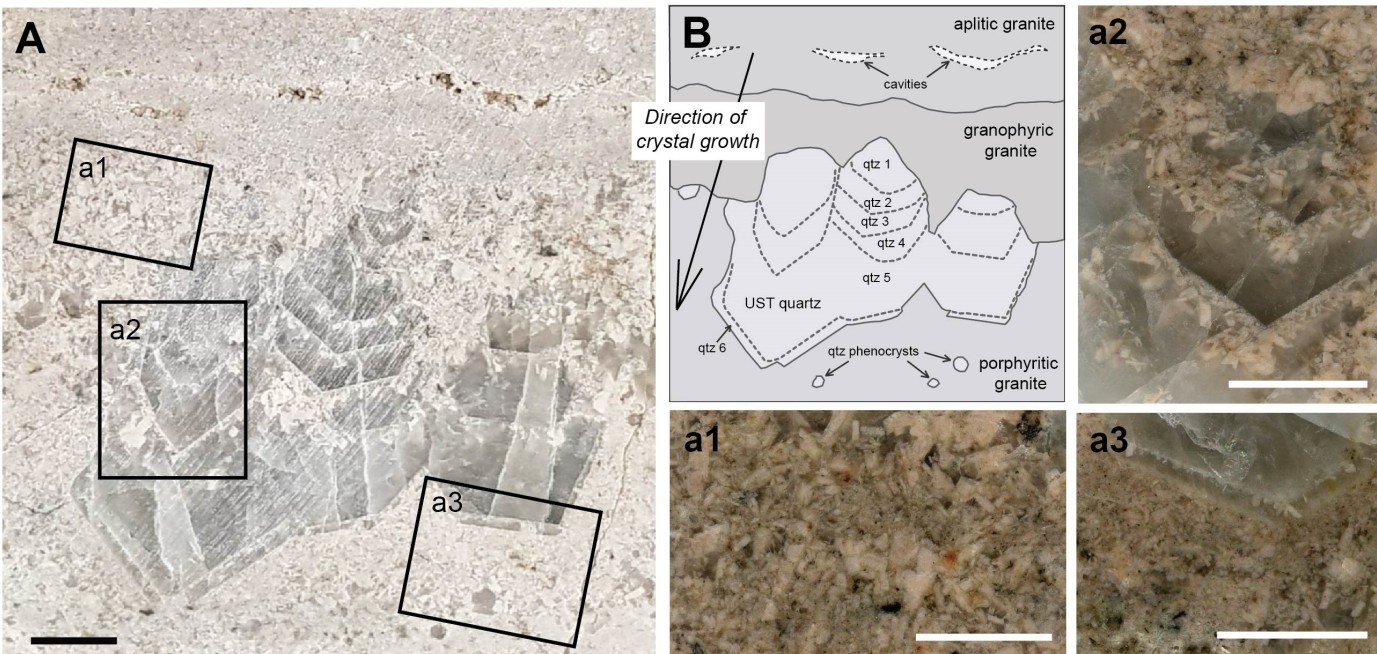

**Figure 2.** (**A**) Photo and (**B**) sketch of rock slab showing a sequence of the Eonyang UST samples. Scale bars represent 2 cm. The UST sample shows a temporal sequence of igneous and magmatic-hydrothermal textures, including the earliest aplitic granite with sub-horizontal cavities, (**a1**) granophyric granite showing the co-growth of feldspar-quartz, (**a2**) UST quartz showing the growth of larger crystals of feldspars and quartz, and (**a3**) the latest quartz porphyry. The UST quartz develops a clear perpendicular zonation along its c-axis showing the six stages of quartz generations (qtz 1–qtz 6). Feldspar in the UST sample is partly altered to sericite.

## 2. Geological Background

### 2.1. Gyeongsang Basin and Cretaceous Granitoids

The Cretaceous Gyeongsang basin, located in the southeastern part of the Korean Peninsula, consists of non-marine sedimentary sequences and volcanic-volcaniclastic rocks, collectively called the Gyeongsang Supergroup, which was deposited above a Precambrian metamorphic basement (Yeongnam massif). The Gyeongsang Supergroup is divided into sedimentary groups, including the oldest Hayang group in the northwest, the Sindong group in the center, and the youngest Yucheon group in the southeast. The Yucheon group is a middle Cretaceous volcanic-volcaniclastic rock sequence consisting of rhyolitic and dacitic volcanic rocks and tuffaceous volcaniclastic rocks, which have an age of approximately 80–100 Ma [33]. Late Cretaceous granitoids with ages of approximately 67–76 Ma (Group IV in Kim et al., 2016) intrude the Gyeongsang Supergroup. The granitoids range from granodioritic to granitic composition, contain magnetite, and the major elements in the rock suggest that they are I-type and high-K calc-alkaline [34]. The granitic magma is associated with late Cretaceous arc magmatism and has compositions reflecting geochemical mixtures of arc magma and continental crust [34].

### 2.2. Sannae-Eonyang Granite

The Sannae-Eonyang granite is one of the Late Cretaceous granitoids [24], which intruded the Yucheon volcanic group of the Gyeongsang Supergroup (Figure 1). The eastern part of the granite is in contact with the earlier Sindong group sedimentary rocks, including dark-green shale-silt-sandstone by later lateral offset by the Cenozoic Yangsan transform fault [35]. The Sannae-Eonyang granite comprises biotite-bearing granitoids that include a series of quartz-monzodiorite and granodiorite intrusions with equigranular, seriate, and porphyritic textures. Granite is the major phase among the intrusions, dated approximately 73 Ma by zircon U-Pb age [25]. They are formed by I-type and high-K

calc-alkaline magma [25,36,37]. The granite often contains mafic enclaves and has aplitic-pegmatitic textures in several portions with numerous miarolitic cavities.

### 2.3. Sannae Deposit

The Sannae deposit is a quartz vein-type W-Mo deposit hosted in stocks of quartz-monzonite and granite in the western part of the granite (Figure 1). The ore grades of $WO_3$ and $MoS_2$ are approximately 2.0 wt.% and 1.0 wt.%, respectively, with molybdenite, wolframite, and scheelite as major ore minerals and quartz-muscovite as a major alteration assemblage [27]. The W-rich zone in the Sannae deposit is closely overlapping with, but relatively shallower than, the Mo-rich zone, and the molybdenite deposition is texturally followed by wolframite-scheelite deposition [27]. Aqueous liquid-rich fluid inclusions with salinities of approximately 10–20 wt.% (NaCl equiv.) predominate in the quartz veins of the deposit [27,28]. The paragenetic sequence of early Mo and later W-Mo coincides with decreasing homogenization temperatures of the fluid inclusions, 350–550 °C during molybdenite deposition and 300–450 °C during wolframite-scheelite deposition [27]. Post-mineralization amethyst quartz often occurs in later vugs [27].

### 2.4. Miarolitic Cavities and Eonyang Amethyst

Numerous miarolitic cavities occur in the aplite or pegmatite in the Sannae-Eonyang granite. Amethyst (known as Eonyang amethyst) occurs in cavities approximately 1 m in diameter in the eastern part of the granite (Figure 1). The occurrence of miarolitic cavities indicates that the granite is a volatile-saturated, relatively shallow emplaced intrusion. Fluid inclusions in the quartz crystals of miarolitic cavities, including the Eonyang amethyst [28–32,38], demonstrate that aqueous phase fluid inclusions predominate, whereas some aqueous fluids are mixed with carbonic vapors. Among the aqueous inclusions, liquid-rich inclusions with bubble sizes of approximately 10–20 vol.% are the most common type of fluid inclusion. They have salinities of approximately 15–20 wt.% (NaCl equiv.) with variable homogenization temperatures of 150–400 °C. Hypersaline brine and vapor-rich inclusions are common types of fluid inclusions. The salinities of the brine inclusions are approximately 30–40 wt.%, and the homogenization temperatures are approximately 300–400 °C. The occurrence of brine and vapor-rich phases indicate that phase separation from a single-phase aqueous fluid occurs above approximately 600 bars [28,29,31]. Amethyst-bearing crystals in the Eonyang amethyst deposit consist of earlier colorless quartz and a later overgrowth of amethyst [29,38]. Fluid inclusions and oxygen stable isotope studies have suggested that the early transparent quartz was formed by a magmatic-dominant fluid, whereas the later amethyst was formed by an aqueous fluid possibly derived from a meteoric source [29]. The aplite or aplite-pegmatite granite associated with the Eonyang amethyst-bearing cavities contain layers of comb-shaped quartz crystals with unidirectional solidification textures (UST). The quartz crystals in the phenocrysts and a clear-zoned UST contain assemblages of melt and fluid inclusions suitable for studying the temporal magmatic-hydrothermal evolution during UST formation and potential W-Mo mineralization.

## 3. Samples and Inclusion Petrography

### 3.1. UST Rock Texture

In the Eonyang UST sample, feldspar (from sub-mm up to 1 cm) and quartz crystals with variable sizes (from sub-mm up to 5 cm) were the major minerals (Figure 2), while accessory minerals such as disseminated magnetite and rutile also occurred. The rim of the feldspar in the sample was partly altered to sericite. The UST sample studied showed a zonal sequence of the following four types of igneous textures from top to bottom: (1) cavity-bearing aplite, (2) granophyric granite, (3) zoned UST quartz crystals, and (4) porphyry with quartz phenocrysts (Figure 2). The sequence of the four textures was horizontally aligned perpendicular to the c-axis of the UST quartz crystals, and the UST sequence was sub-horizontally followed by another sequence of downward UST. The

aplite with cavities had the smallest grain size, with the cavities elongated perpendicular to the c-axis of the UST quartz crystal. The grain size of the granophyric texture granite was larger than that of the aplite and showed a co-crystallization texture of feldspar and quartz. The UST quartz consisted of several clear zones of quartz-feldspar crystals growing in the direction of the c-axis of the quartz crystal, and each zone in the UST quartz started with the crystallization of euhedral feldspar crystals followed by quartz-only crystallization. The sequential crystallization patterns in each zone in the UST quartz were repeated approximately 3–6 times until the rock textures changed to a porphyritic rock containing quartz phenocrysts (Figure 2).

### 3.2. Melt and Fluid Inclusions in Quartz Phenocryst and UST Quartz

Fluid and melt inclusions occurred in both quartz phenocrysts and UST quartz. Based on the phase proportions in the fluid inclusion, we classified the fluid inclusion types as follows: (1) aqueous liquid-rich inclusions, (2) aqueous vapor-rich inclusions, and (3) aqueous salt-bearing hypersaline brine inclusions (Figure 3). The bubble sizes of the liquid-rich, brine, and vapor-rich inclusions were approximately 20–30, 10, and 80–90 vol.%, respectively. All inclusion types occurred in the quartz zones of the UST and quartz phenocrysts in the porphyritic granite. Most fluid inclusions in the sample were textually grouped or lay along linear trails, and therefore were assigned as a cogenetic fluid inclusion assemblage (FIA) [39,40]. We studied a petrographically defined pseudo-secondary FIA based on the texture of the quartz zones and the FI trails. Two types of FIAs were common in the USTs and quartz phenocrysts: (1) an assemblage of liquid-rich inclusions and (2) an assemblage of brine inclusions. All the FIA types are distributed with a similar population among the studied quartz. We also found an assemblage of coexisting brine and vapor-rich inclusions (i.e., a "boiling assemblage"; Figure 3E).

Silicate melt inclusions (SMIs) occurred in quartz phenocrysts in the quartz porphyry and were observed to be internally crystallized (Figure 3F). SMIs also occur in the earlier zones (zones 1 and 2) of the UST quartz. SMIs in the quartz phenocrysts were not texturally associated with fluid inclusions (Figure 3F); however, SMIs in the UST quartz occur within pseudo-secondary trails consisting of coexisting melt and aqueous liquid-rich inclusions (Figure 3B). Some individual inclusions were internally mixed with phases of silicates and aqueous fluids (Figure 3A).

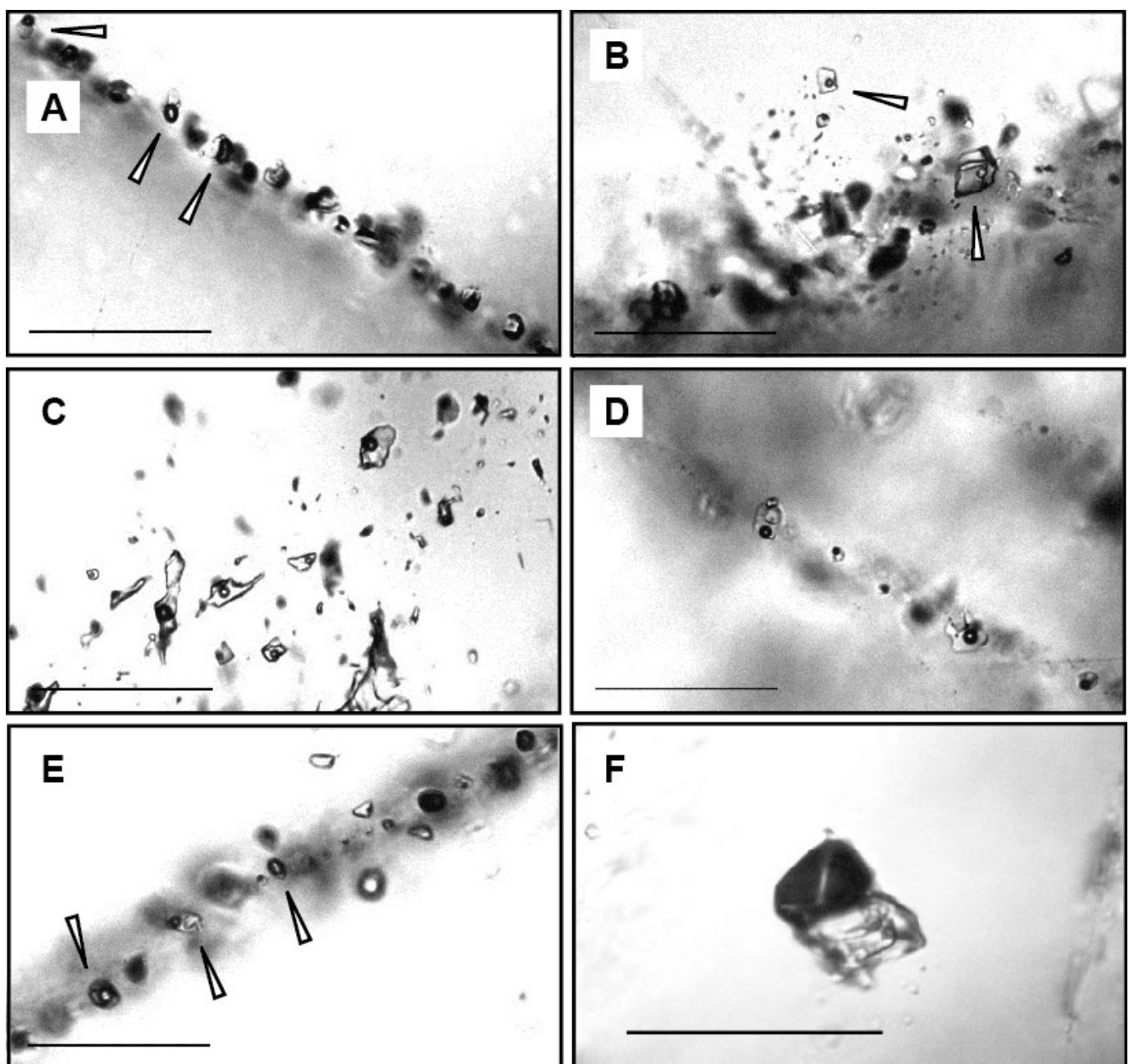

**Figure 3.** Photomicrographs of fluid and melt inclusions in the Eonyang UST sample. Scale bars represent 50 μm. (**A**) A pseudo-secondary assemblage of melt-fluid inclusions in the UST quartz. The arrows show inclusions of mixed silicate melt and aqueous fluid. (**B**) A pseudo-secondary assemblage of silicate melt inclusions and aqueous fluid (liquid-rich) inclusions in the UST quartz. The melt and fluids (arrowed) are possibly trapped co-genetically. (**C**) A pseudo-secondary assemblage of aqueous liquid-rich inclusions in the UST quartz. (**D**) A pseudo-secondary assemblage of hypersaline (brine) inclusions in the UST quartz. (**E**) A pseudo-secondary assemblage of co-trapped brine and vapor-rich inclusions (boiling assemblage) in the UST quartz. (**F**) A silicate melt inclusion in a quartz phenocryst. The melt inclusion is internally crystallized.

## 4. Methods

### 4.1. Microthermometry

Microthermometry of fluid inclusions was performed using a Linkam FTIR 600 (similar to THMS 600) heating-cooling stage at Inha University, Incheon, Korea. The stage was calibrated based on three temperatures using synthetic fluid inclusions: the triple point ($-57.1$ °C) of the $CO_2$-Ar mixed aqueous inclusion, and the ice melting (0.0 °C) and critical homogenization (374.0 °C) of a pure $H_2O$ inclusion. A total of 14 double-polished

quartz chips from the UST stage quartz and phenocrysts were prepared (Tables S1 and S2). Three to five single fluid inclusions per FIA were analyzed using microthermometry. Assemblages of liquid-rich and brine inclusions were analyzed, where brine inclusions were associated with brine-only or boiling assemblages (Table S1).

### 4.2. Laser Raman Spectroscopy

In situ laser Raman spectroscopy was performed on the bubble part of the liquid-rich and vapor-rich inclusions to detect gaseous species such as $CO_2$ and $CH_4$ in the studied fluid inclusions. We used a LabRam Hr Evolution (Horiba) confocal microscope at Inha University. A total of 11 double-polished chips were selected on six quartz stages in the UST and quartz phenocrysts. A total of 101 single liquid-rich and vapor-rich inclusions were analyzed.

The equipment was set up with an 1800 g/mm grating and allowed to reach diffraction levels in the range of 450–800 nm. The laser was equipped with a 532 nm edge filter with a spot size <2 μm. The signal acquisition and accumulation times were adjusted to 10 s to optimize the signal-to-noise ratio. The analysis was performed exclusively on the inclusions near the surface to obtain better signals.

### 4.3. LA-ICP-MS

LA-ICP-MS microanalysis of brine inclusions was conducted after microthermometry (Table S2). LA-ICP-MS analysis of brine and melt inclusions hosted in the UST quartz and phenocrysts was conducted using an LA-ICP-MS system at ETH Zurich, Zurich, Switzerland. The GeoLas 193 nm ArF Excimer laser system was connected to a quadrupole ICP-MS (Perkin Elmer NexION 2000). The flow rate of the He carrier gas in the ablation chamber was 1 L/min. maintaining an oxide production rate ($ThO^+/Th^+$) under 0.5%. The energy density of approximately 20 J/cm$^2$ and repetition rate of 10 Hz were applied to the inclusions hosted in quartz. Using an iris aperture, the laser spot size was gradually increased for a complete excavation of the inclusion material without fracturing the hosting quartz crystal. The following isotopes were chosen to analyze with a dwell time of 2–20 ms: $^7$Li, $^{11}$B, $^{23}$Na, $^{32}$S, $^{35}$Cl, $^{39}$K, $^{55}$Mn, $^{57}$Fe, $^{63}$Cu, $^{66}$Zn, $^{71}$Ga, $^{73}$Ge, $^{75}$As, $^{79}$Br, $^{85}$Rb, $^{88}$Sr, $^{89}$Y, $^{90}$Zr, $^{93}$Nb, $^{95}$Mo, $^{107}$Ag, $^{111}$Cd, $^{115}$In, $^{118}$Sn, $^{121}$Sb, $^{133}$Cs, $^{137}$Ba, $^{139}$La, $^{140}$Ce, $^{182}$W, $^{205}$Tl, $^{208}$Pb, $^{209}$Bi, $^{232}$Th, and $^{238}$U.

For external standardization to calculate the element ratios, we used a standard reference material (SRM) of a fused synthetic glass of NIST 610 [41] and a natural scapolite mineral of Sca-17 [42,43] for S, Cl, and Br. For internal standardization, we used an estimated Na concentration from microthermometry [44] for the brine inclusions and an average $Al_2O_3$ content of $14 \pm 1$ wt.% in the Sannae-Eonyang granite [26,36,37] for the melt inclusions. Because Al concentrations in magma might change during the late stage of magma crystallization, for melt inclusion analysis, only the element ratios are discussed in the present study. The concentrations of trace elements in the UST and phenocryst quartz were simultaneously (Table S3) obtained during the LA-ICP-MS analysis of inclusions. We applied a stoichiometric value of Si to quartz as an internal standard. To calculate the quartz composition, LA-ICP-MS transient signals were carefully observed not to incorporate any signal of inclusions. Data reductions and quantifications from the LA-ICP-MS transient signals were performed using the SILLS software suite [45].

## 5. Results

### 5.1. Microthermometry

Salinities, homogenization temperatures, and densities of the liquid-rich inclusion assemblages were highly variable in the UST quartz compared to the postdating phenocrysts (Table S1 and Figures 4 and 5). The homogenization temperature of liquid inclusions represents the temperature at which the vapor bubble disappears (L + V → L: L = liquid and V = vapor bubble). The ice melting temperatures ranged from −3 to −21 °C in the UST quartz and −11 to −21 °C in the phenocryst, which corresponded to an apparent

salinity [46] from 4 to 23 wt.% NaCl equiv. in the UST quartz and 15 to 23 wt.% in the phenocrysts. Clathrate was not observed in the liquid-rich inclusions during cooling and ice melting experiments. $T_h$ and calculated densities [47] were approximately 170–380 °C and 0.7–1.0 g/cm$^3$ in the UST quartz and 240–320 °C and 0.8–1.0 g/cm$^3$ in the phenocrysts, respectively. In the assemblages in the quartz stage from 1 to 6, we found many fluid densities (Table S1 and Figure 5B).

Brine inclusions are homogenized either via vapor bubble disappearance (L + V → L) or salt dissolution (L + H → L: H = halite or salt crystal) (Table S1). The temperature of L + V → L in the brine assemblages in the UST quartz was 210–380 °C, and that of the phenocrysts was 310–430 °C. The salt melting temperatures and calculated salinities [46] were approximately 230–440 °C and 33–52 wt.% NaCl equiv. in the UST quartz, and 370–470 °C and 44–56 wt.% NaCl equivalent in the phenocrysts, respectively. If the L + V → L homogenization temperature was higher than the salt melting temperature, we calculated the pressure based on the model NaCl-H$_2$O system [47]. Because the salt melting temperatures in many brine assemblages exceeded the temperature of bubble disappearance (L + V + H → L + H), we calculated fluid pressures by liquidus of NaCl [48]. The calculated pressures were highly variable at 80–2300 bars in the UST quartz compared to 640–770 bars in the phenocrysts (Table S1 and Figures 4 and 5). The total homogenization temperatures in the brine assemblages were approximately 320–440 °C in the UST and 370–470 °C in the quartz phenocrysts. We also obtained a large variety of fluid densities in the brine assemblages in quartz stage 4 (Figure 5B).

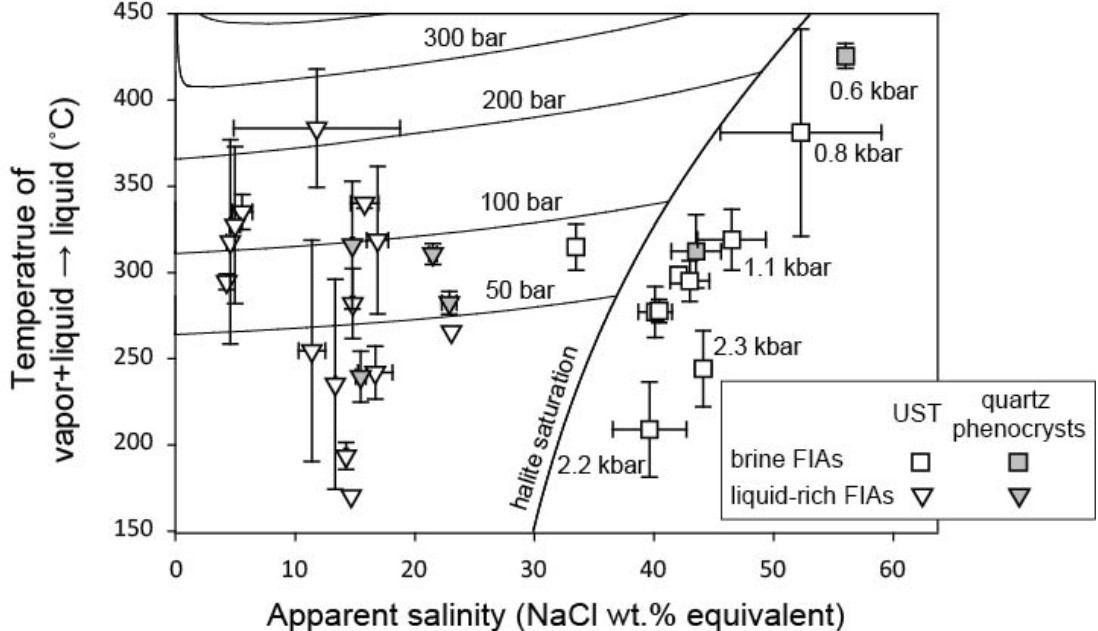

**Figure 4.** Plots of liquid-rich and brine inclusion assemblages showing apparent salinities (wt.% NaCl equivalent) and homogenization temperatures (°C) of vapor bubble + liquid into liquid. Isobaric and halite saturation curves were calculated from a model NaCl-H$_2$O system [47]. Many of the brine inclusion assemblages plotted in the field of halite saturation (totally homogenized by salt dissolution) and brine pressure were calculated with NaCl liquidus [48]. FIAs hosted in the UST are expressed with empty symbols.

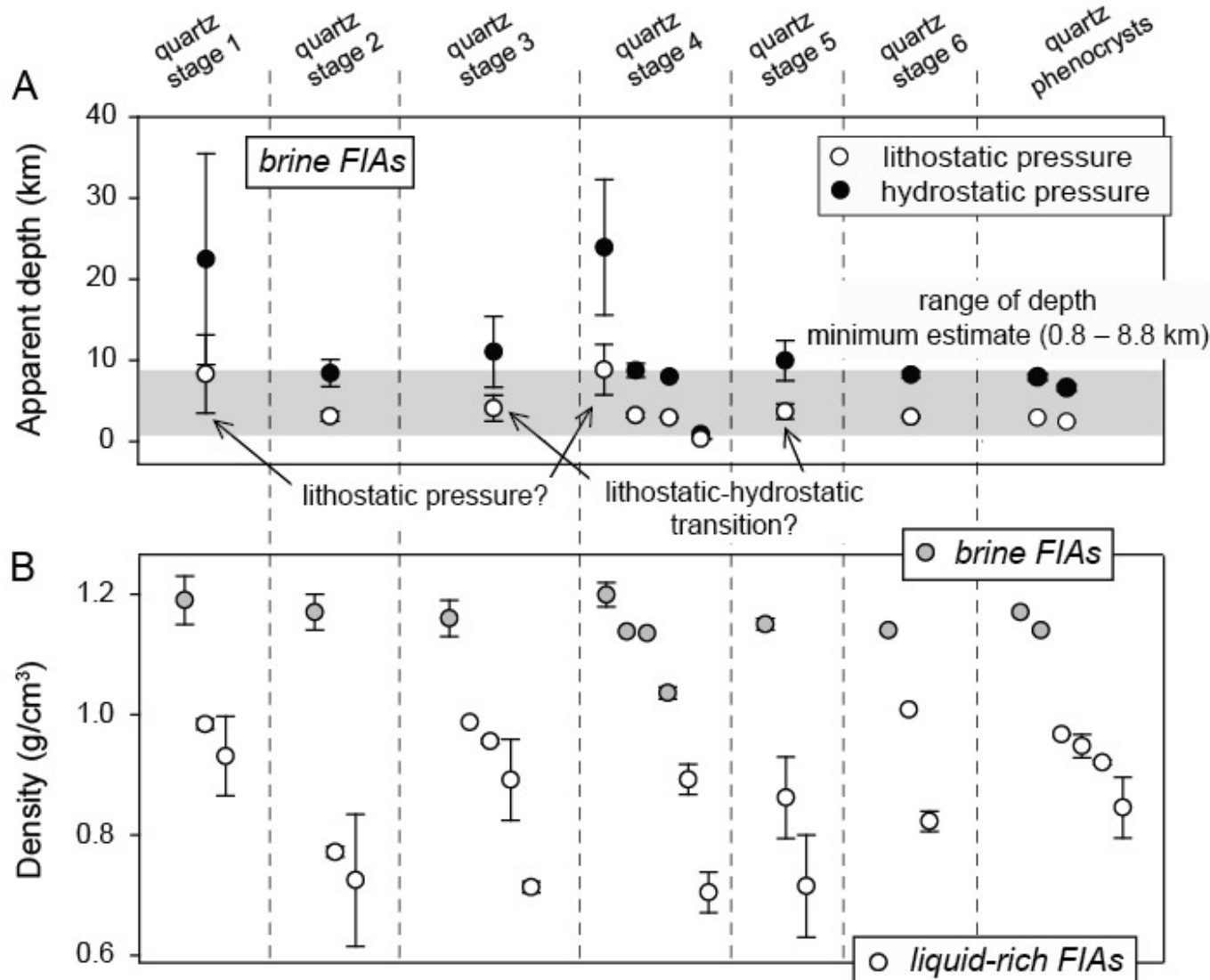

**Figure 5.** Sequences of (**A**) calculated apparent paleodepths from brine inclusion assemblages and (**B**) calculated densities of liquid-rich and brine fluid inclusions assemblages (FIA) in early UST quartz stages (1–6) and postdating quartz phenocrysts. (**A**) We assumed densities of 2.7 and 1 g/cm³ to calculate lithostatic and hydrostatic pressures, respectively. Gray shaded area represents an estimated range of minimum apparent depth, which is more than 1–8 km if we assume the highest pressure (2300 bars) as a lithostatic, and the lowest pressure (80 bars) as a hydrostatic. (**B**) Densities of the brine and liquid-rich FIAs were calculated from salinities and homogenization temperatures (vapor + liquid → liquid) using a model NaCl-H$_2$O system [47].

### 5.2. Raman Spectroscopy

Among the 101 analyses, CO$_2$ peaks at 1285 cm$^{-1}$ and 1388 cm$^{-1}$ were detected in 22 vapor-rich and liquid-rich inclusions in the UST quartz. Although there were arbitrary units in Raman intensities, the CO$_2$ peaks relative to the background noise were higher in the vapor inclusions than in the liquid-rich inclusions (Figure 6). We found peaks of neither CH$_4$ (2917 cm$^{-1}$) nor N$_2$ (2326 cm$^{-1}$) in the fluid inclusions.

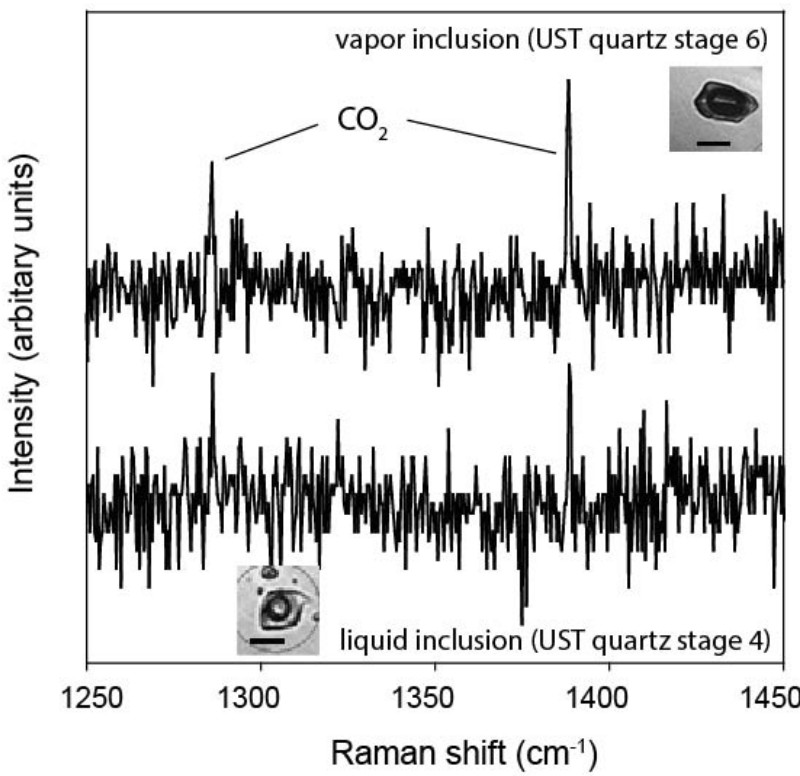

**Figure 6.** Raman spectra of vapor-rich and liquid-rich inclusions in the UST quartz showing $CO_2$ peaks. The Raman spectrometer laser was focused on the vapor bubble part of the inclusions.

*5.3. LA-ICP-MS of Fluid and Melt Inclusions and Hosting Quartz*

Average and standard deviation values were calculated using more than three single-targeted brine inclusion analyses from FIAs (Table S2). Elements such as Na, Li, B, S, Cl, K, Mn, Fe, Cu, Zn, Ga, Ge, As, Br, Rb, Sr, Y, Mo, Ag, Cd, In, Sn, Sb, Cs, Ba, La, Ce, W, Tl, Pb, Bi, Th, and U were detected in the brine inclusions (Table S2 and Figure S1). Although the elemental concentrations in the brine assemblages in the UST quartz and postdating phenocrysts were similar, some elemental concentration levels of B, Fe, Sr, Mo, Cs, W, and U were significantly different (Figure 7 and Figure S1). Fe, Mo, W, and U were higher in UST, whereas B, Sr, and Cs were higher in phenocrysts.

Trace elements in brine inclusions analyzed with LA-ICP-MS can be more accurately expressed using element ratios because they are not affected by an internal standardization based on an assumed fluid inclusion salinity [44]. Although we report absolute concentrations in Table S2, here we report and discuss the elements in the figures by their ratios, such as element/Na, which is independent of the effects of fluid salinity and phase separation [3,49], or Rb/Sr, Mn/Fe, and Cl/Br, which are useful for understanding the geochemistry of magmatic-hydrothermal fluids (Figure 7). The element ratios among the brine assemblages in each stage of the UST quartz were not variable; however, some ratios such as K/Na, B/Na, Cs/Na, W/Na, Mo/Na, Rb/Sr, and Mn/Fe showed a significant deviation between the UST quartz and phenocrysts (Figure 7 and Figure S1). K/Na, Rb/Sr, W/Na, and Mo/Na ratios were higher in the UST quartz, whereas B/Na, Cs/Na, and Mn/Fe ratios were higher in the phenocrysts (Figure 7). The ranges of Cl/Br (ppm) ratios were 510–900 higher than seawater at approximately 290–300 (Figure 7C).

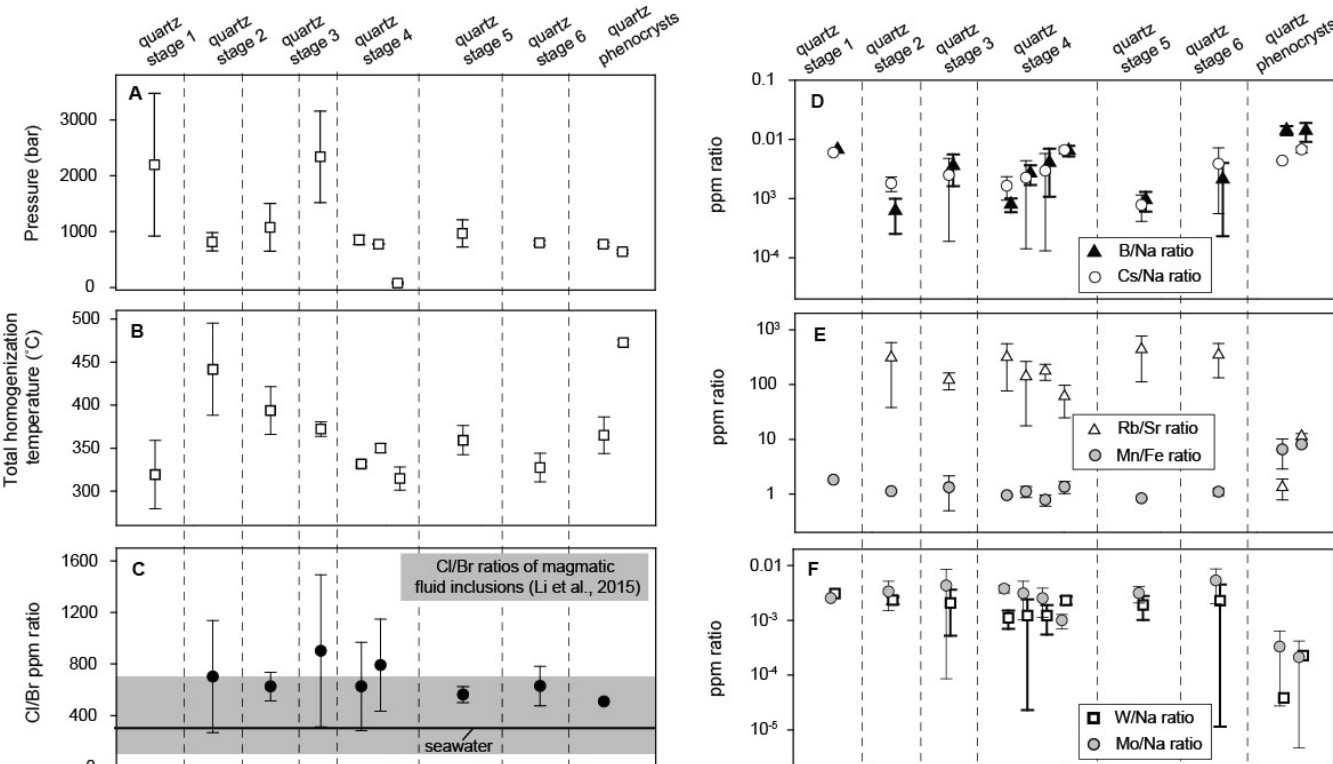

**Figure 7.** Sequences of microthermometric P-T information and trace element ratios (obtained by LA-IC-MS microanalysis) of brine inclusion assemblages in early UST quartz stages (1–6) and postdating quartz phenocrysts. (**A**) Calculated pressures and (**B**) temperatures of total homogenizations in the brine assemblages. Trace element ratios of (**C**) Cl/Br, (**D**) B/Na and Cs/Na, (**E**) Rb/Sr and Mn/Fe, and (**F**) W/Na and Mo/Na in the same brine inclusion assemblages.

The compositions of the three melt inclusions hosted in the quartz phenocrysts are listed in Table S2. Because the aplitic porphyry is highly altered after UST formation and porphyry solidification, many of the elements might have been later remobilized and modified. We selected and used ratios of elements including Mg, Al, Nb, and Ti, which might not be strongly fluid mobile [50,51].

Elements such as Li, Mg, Al, Na, P, K, Ti, Mn, Fe, Ga, and Ge were detected in most of the quartz matrix, whereas B, Mg, Mn, Fe, Y, Zr, In, and Sn were detected only in some quartz. Because alkali elements such as Li, Na, and K might be incorporated from fluid inclusions into the quartz signal, we do not discuss them further. Al and Ti were dominant among the trace elements (Figure 8). The Al and Ti concentrations in the UST quartz were significantly higher than those in the phenocryst quartz. Al and Ti were 110–240 and 80–130 µg/g in the UST quartz and 70–80 and 70–80 µg/g in the phenocrysts, respectively (Table S3). The Al and Ti concentrations in the UST quartz gradually increased from quartz stages 1 to 6 (Figure 8). The ranges of B, P, Ga, and Ge concentrations in the quartz of the UST and phenocrysts overlapped.

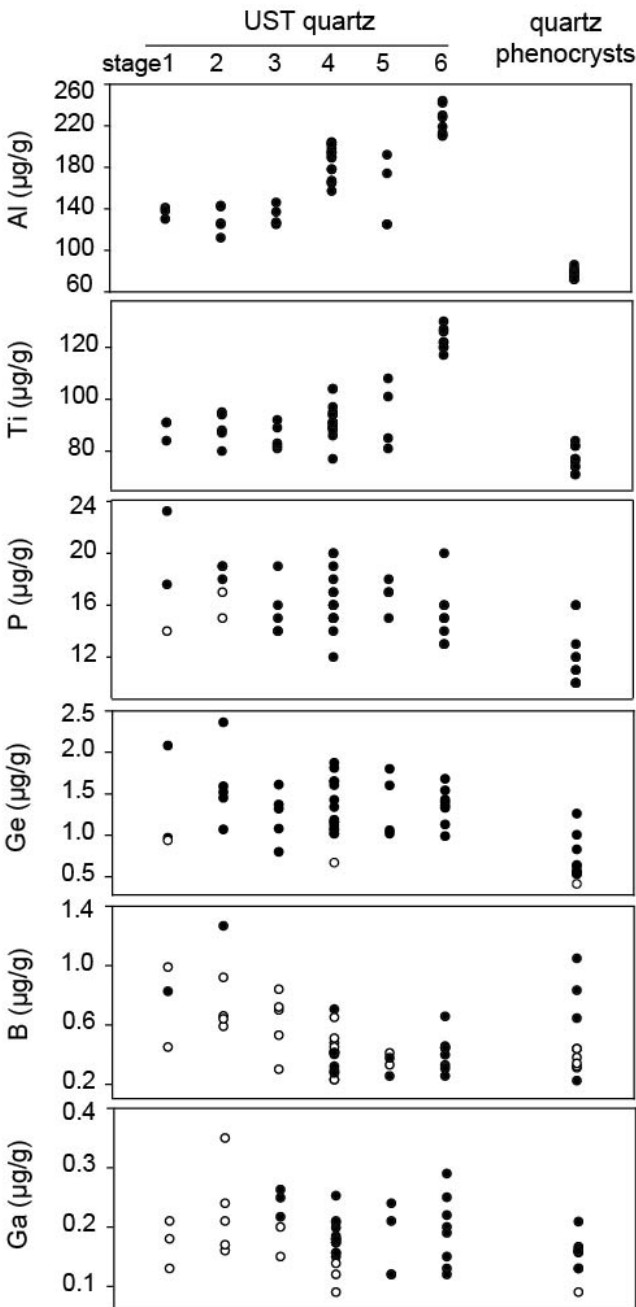

**Figure 8.** Trace element compositions in quartz in UST quartz stages and quartz phenocrysts. Empty symbols represent detection limits of LA-ICP-MS calculated by the protocols described by Longerich et al. (1996) [52] adopted in the SILLS program [45].

## 6. Discussion

### 6.1. Pressure Conditions and Emplacement Depths

The homogenization temperature ($T_h$) and calculated P from the microthermometry of liquid-rich and brine inclusions indicated a minimum estimation of T and P. For brine inclusions, $T_h$ represents a total homogenization, and it can be a temperature of either L + V → L or L + H → L (see microthermometry section in Results). The $T_h$ and salinity obtained from microthermometry allowed the calculation of the densities of fluid inclusions and comparison with the isodensity P-T curves (isochores). To obtain an actual P-T at fluid entrapment, we needed an additional correction of an independently obtained T or P.

In the UST quartz, $T_h$ and calculated pressures from brine inclusion assemblages were 320–440 °C and 80–2300 bars, respectively. Paleodepths can be estimated based

on the hydrostatic (density of 1 g/cm$^3$) or lithostatic (density of 2.7 g/cm$^3$) pressure regimes. The calculated depths from the microthermometry of brine assemblages were 0.8–23.9 km at hydrostatic pressure and 0.3–8.8 km at lithostatic pressure (Figure 5A). Because such a large range of depths is not naturally realistic during UST formation, the large variations in the fluid pressures might be due to the switching of the pressure regimes. If we assume two brine assemblages estimating exceptionally high pressure (2.2–2.3 kbar produced by lithostatic conditions) and one brine assemblage estimating the lowest pressures (80 bar produced by hydrostatic conditions), we could obtain a minimum estimation of the paleodepth of 1–9 km during the Eonyang UST formation (Figure 5A). The other six brine assemblages, estimating an intermediate range of pressures of 800–1100 bars, might represent a fluid transition state in the pressure regimes. Because the hydrothermal pressures and apparent depths estimated a minimum value, the Eonyang UST might have been formed at a depth of more than 8.8 km.

### 6.2. Melt Inclusion and Magma Fractionation

USTs have been globally recognized in highly fractionated granitoids [5,7,9,19]. The CIPW normative mineralogy [53] of the Sannae-Eonyang granitoids was calculated from its major elemental compositions [26,36,37] and plotted in a quartz–alkali-feldspar–plagioclase ternary diagram with experimental eutectic and cotectic information [54–56] (Figure 9). Elemental ratios of MgO/Al$_2$O$_3$ and Nb/TiO$_2$ obtained from melt inclusions in the phenocrysts were plotted and compared to the ratios in the Sannae-Eonyang granitoids (Figure S2). A pressure of 80–2300 bars obtained from the brine assemblages (Figure 5A) and elemental ratios in the melt inclusions (Figure S2) indicated that the Eonyang UST rock was formed at a relatively fractionated stage of the Sannae-Eonyang granite crystallization. The MgO/Al$_2$O$_3$ and Nb/TiO$_2$ ratios in the melt inclusions were compared with the bulk compositions, and the SiO$_2$ contents in the melt inclusions were approximately 76–80 wt.% (Figure S2).

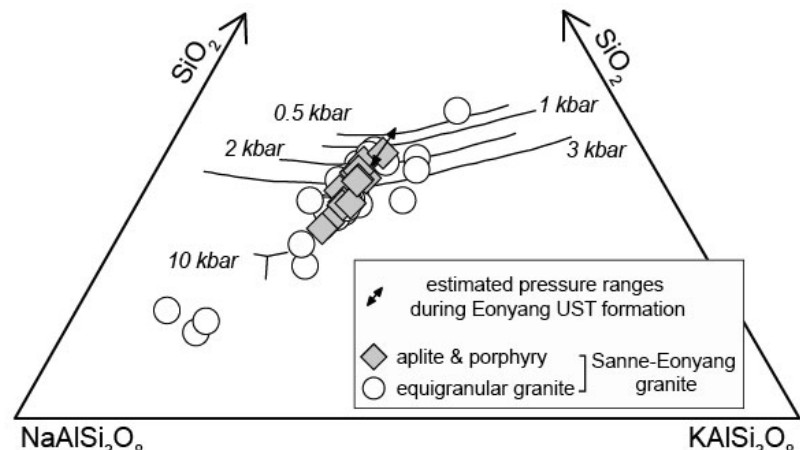

**Figure 9.** Normative mineralogy in the Sannae-Eonyang granite calculated from major element compositions [26,36,37] are plotted in a quartz–alkali-feldspar–plagioclase ternary (QAP) diagram. We applied the eutectic-cotectic information [54–56] and plotted a possible pressure range (820–2200 bars) estimated from the fluid inclusion analysis.

### 6.3. Formation of UST

Liquid-rich inclusions with 20–30 vol.% of a vapor bubble in the Eonyang UST samples are texturally associated with silicate melt inclusions (Figure 3B) and provided strong evidence that the aqueous liquid directly exsolved from silicate magma. Although the information on T$_h$ and the calculated pressure of the brine and liquid-rich inclusions only indicated a minimum hydrothermal P-T, the density of the inclusion could be directly estimated from the model H$_2$O-NaCl system [47], which represented the density of a single fluid released from the magma. One brine assemblage (at UST quartz stage 4), showing the

behavior of earlier salt melting before total homogenization, could be formed as a phase separation of an aqueous intermediate-density fluid vapor [2–4] indicated by the existence of a boiling assemblage (Figure 3E). Highly fluctuating densities in the brine and liquid-rich assemblages were remarkable in each quartz stage in the UST, whereas the density fluctuation in the quartz phenocrysts was not as severe as that in the UST (Figure 5B). Such a specimen-scale density variation can be explained by the fluctuation of hydrothermal pressures caused by a sudden regime change between lithostatic and hydrostatic pressures. The transition of the pressure state affects the salinities of an exsolving aqueous fluid from silicate magma [57–59], producing extremely different salinities in the liquid inclusions of 4–23 wt.% and brine inclusions of 34–52 wt.% in the UST quartz. This observation was consistent with a previous estimation [60] in which lithostatic hydrothermal fluid connected to an overlying fracture system became periodically hydrostatic. In the Eonyang UST (Figure 5A), for example, the liquid-rich assemblages had higher lithostatic pressures that were dominant in stage 1 quartz relative to stage 2 quartz, which had relatively lower hydrostatic pressures. A highly fluctuating fluid pressure was estimated from the brine inclusion assemblages; for example, $2200 \pm 1300$ bars in stage 1 quartz and $820 \pm 160$ bars in stage 2 quartz were calculated in the UST quartz. The 2.2 kb in stage 1 quartz could be caused by a lithostatic pressure at approximately 8 km depth, and the 0.8 kb in stage 2 quartz could cause hydrostatic or hydrostatic-lithostatic transitions at a similar apparent depth below the paleosurface. These highly variable pressure estimates from the brine inclusions in the Eonyang UST quartz strongly suggest a transition of lithostatic-hydrostatic hydrothermal pressure regimes and that it might be causative for UST formation.

This highly transient magmatic-hydrothermal pressure causes the formation of igneous textures [5,9,21] in the volatile-rich igneous rock [22] by shifting the cotectic boundary of feldspar and quartz (Figure 9). Lithostatically-formed high hydrothermal pressure up to approximately 2 kb during the early stage of the Eonyang UST was possibly caused by a large amount of fluid released from the upper part of the crystallizing magma of the Sannae-Eonyang granite. Such an elevated lithostatic pressure shifted the cotectic boundary from a field of coexisting feldspar-quartz into a field of only quartz (Figure 9), which was consistent with the petrographic observation of the Eonyang UST from the granophyric granite into the UST quartz (Figure 2B). Fluctuating pressures during UST formation due to a transition to hydrostatic pressure might be caused by the hydrofracturing of rock, as shown by an SEM-CL study of the UST quartz [20], which might be subsequently sealed by a new generation of quartz that then created an elevated lithostatic pressure. Such highly fluctuating, but also highly-lithostatic, hydrothermal pressures continued to form larger crystals of early feldspar and late quartz in the UST sequence, possibly until the cessation of a single batch of magmatic fluid release. After the release of a large amount of magmatic fluid, a shifting of liquidus occurred, and it would "quench" an undercooled silicate melt to a fine-grained aplite or quartz porphyry [61,62], which was consistent with the petrography of the Eonyang UST sample (Figure 2B).

In quartz phenocrysts, brine assemblages showed relatively lower minimum pressures of approximately 640–770 bars (minimum apparent depths of approximately 7–8 km of corresponding hydrostatic pressures), suggesting that the fluids were at relatively low hydrostatic pressure after UST and quartz porphyry formation. From microthermometry in the Eonyang UST, the periodic release of a large amount of magmatic fluid might have created a single UST sequence in the upper part of the crystallizing granite. Because the density of $CO_2$ is lower than that of $H_2O$, the presence of $CO_2$ mixed in the aqueous fluids (Figure 6) might promote the pressure transition during magmatic fluid release, although it cannot be quantitatively investigated.

Ti and Al are the two dominant trace elements substituted in the quartz structure that record the geochemical conditions of the system during quartz generation [63–65]. In a $TiO_2$-saturated (rutile-bearing) system, Ti concentrations in quartz show a positive correlation with temperature but a negative correlation with pressure [66]. Al concentrations in hydrothermal quartz are dependent on temperature [67,68] and Al solubilities in the fluids,

which are pH-dependent [64]. Müller and Koch-Müller (2009) [69] and Was (2017) [68] suggested that $Al^{3+}$ substitution in quartz can be coupled with $H^+$ at the site of $Si^{4+}$ in quartz, which might imply that decreasing the pH could promote Al substitution in quartz. Müller et al. (2002) suggested that a high Al content in hydrothermal quartz represented a faster growth rate of quartz. In the Eonyang UST sample, we cannot apply Ti-in-quartz geothermometry because of its high kinetic affinities, preventing it from attaining equilibrium [66]. The gradual increase of Ti and Al in stage 1 to stage 6 quartz (Figure 8) can rather be related to an increasing quartz growth rate [66,70] during UST formation. High Ti contents (>100 ppm) in rapidly grown hydrothermal quartz have also been observed in porphyry deposits [71].

### 6.4. Magmatic Fluid Exsolution and P-T Evolution during the Eonyang UST Formation

Temporal P-T changes in the magmatic-hydrothermal fluids were preserved in the fluid inclusions in the Eonyang UST (Figures 5 and 7). Transient lithostatic-hydrostatic pressure was suggested to be a cause for the various densities and salinities in the liquid-rich and brine inclusions in the UST quartz (Figure 10). Because some of the fluid inclusions were associated with SMIs (Figure 3B), some isochores might be expected to cross the solidus of fluid-saturated granite [72]. The isochore of assemblages of the lowest density (approximately 0.70 g/cm$^3$) intersected the granite solidus at approximately 2.5 kb (Figure 10), which was similar to the two maximum "lithostatic" pressures (2.2–2.3 kb) estimated from the brine inclusion assemblages. Therefore, the magma exsolved some portion of the liquid phase fluids at a minimum pressure of 2.5 kb and a maximum corresponding apparent paleodepth of 10 km. The magmatic fluid exsolved at elevated pressure promoted preferential crystallization of quartz in the UST sequence. The addition of F, B, and other network-modifying volatiles to the granite system shifted the water-saturated granite eutectic temperature [56,59], which allowed isochores of higher densities to cross the granitic solidus (Figure 10). A transition into a hydrostatic pressure was recorded in some of the brine assemblages, possibly caused by a hydrofracturing, lowered pressure of the magmatic fluid down to 0.1–0.8 kb (Figures 10 and 11). Tosdal and Dilles (2020) suggested that magmatic fluids of lithostatic pressure penetrate the ductile zone above the magma chamber, and when they hit the overlying brittle zone, the fractures move due to the hydrostatic pressure. The oscillation in pressure may accompany a pulse of magmatic fluid release [60].

From the calculated isochores of the brine assemblages in the UST (Figure 10), we derived the possible temperature ranges during the UST formation in Eonyang. The temperature range must be from the $T_h$ (minimum) to the isochoric temperature at the upper continental crust depth (maximum); here, we assumed 10 km. From the brine assemblages of the two highest densities, we obtained temperature ranges of 320–350 °C and 370–390 °C. Although the temperature in the isochoric ranges can extend slightly if we correct it with higher pressure, these fluid temperature ranges would be far below the equilibrium of the water granitic solidus [72]. These low temperatures suggest a significant undercooled disequilibrium during UST formation in granite. Skeletal textures such as granophyric texture and UST occurring in the pegmatitic and aplitic granites are evidence of strong undercooling [73–76].

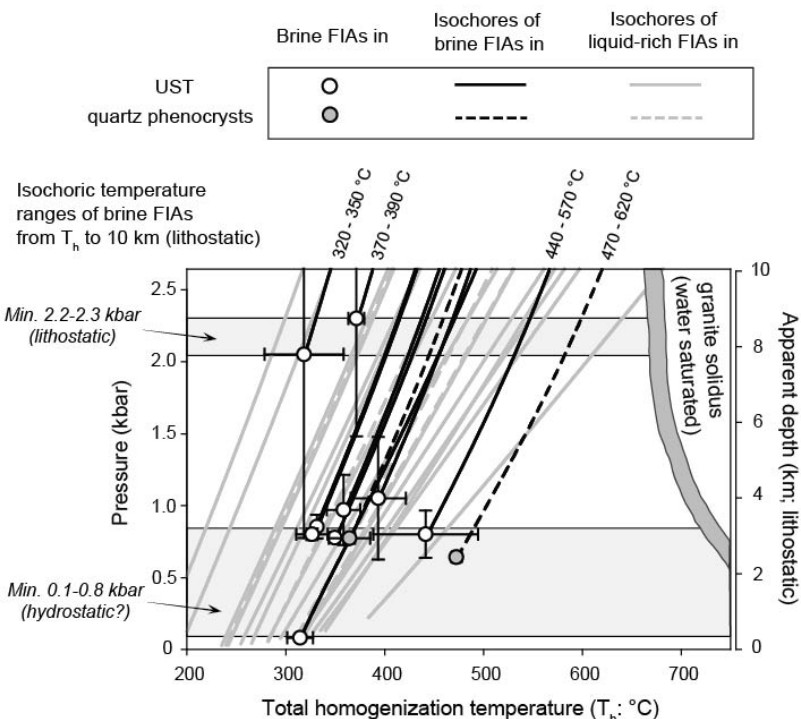

**Figure 10.** Diagram of P-T information obtained from microthermometry of brine inclusion assemblages and isochores of liquid-rich and brine inclusion assemblages in the Eonyang UST sample. Two brine assemblages showing the highest minimum pressures (2.2–2.3 kb) can be assumed to be lithostatic, while other assemblages of lower pressures (0.8–1.1 kb) are hydrostatic or transitions of hydrostatic-lithostatic pressures at the same depth. Aqueous fluids of various salinities exsolved at the lithostatic pressure are expected to experience a transient shift in hydrostatic pressure, possibly by hydrofracturing. Transient pressure drop by the hydrostatic-lithostatic transition might cause further magmatic fluid exsolution. Ranges of isochoric temperatures from total homogenization temperatures ($T_h$) to the temperature of 10 km lithostatic depth obtained from the two highest density brine inclusion assemblages (1.19–1.20 g/cm$^3$) are about 320–350 and 370–390 °C, which are far below the solidus of water-saturated granite [72].

*6.5. Geochemical Evolution of Magmatic-Hydrothermal Fluids*

Although highly fluctuating pressures and temperatures were recorded in the Eonyang UST, we could not find a significant variation in the trace elements in the brine assemblages (expressed as ratios in Figure 7 and Na-normalized ratios in Figure S1). Such relatively constant trace element ratios in the fluids further support the hypothesis that a single batch of magmatic fluid was released during the formation of a single sequence of UST. The constant trace element ratios also implied that the UST was formed from an aqueous fluid-rich melt in a relatively short period. The increasing Al content in the UST stage quartz (Figure 8) suggested a gradual increase in the relative growth rate of quartz crystals in the UST.

Relatively constant trace element ratios in the brine assemblages in the UST were highly contrasted to the ratios in the phenocrysts in the postdating aplite porphyry (Figure 7 and Figure S1). Among the ratios, increasing B/Na, Cs/Na, and Mn/Fe and decreasing Rb/Sr, W/Na, and Mo/Na were noticeable. Increasing B and Cs in magmatic-hydrothermal fluids indicate the effect of fractional crystallization [3,4,23]. The simple fractionation could not explain the Rb/Sr, W-Mo, and Mn/Fe trends. Rb/Sr generally increased during the latest stage of fractionating magma, which was not consistent with the Eonyang UST sample. W and Mo are incompatible elements, which should show a similar increasing trend as that of Cs. The significant decrease in W and Mo concentrations might indicate W-Mo mineralization temporally between the UST and after solidification of the quartz por-

phyry. No economic W-Mo deposits were located or discovered near the Eonyang amethyst mine, although it might indicate a hidden W-Mo orebody nearby. If the decrease in metal concentrations was not caused by mineralization, a new (possibly geochemically primitive) magma, or similar magma without fluid extraction (to account for the decreasing Rb/Sr ratios), was derived after the UST. The Mn/Fe concentrations in rock-buffered magmatic fluids are controlled by their relative redox state and elevated Mn/Fe ratios in relatively oxidized fluids [77]. The Mn/Fe ratios in brine inclusions were tested by early Cu-Au-bearing veins and petrographically Mo-bearing veins in the Bingham Canyon deposit [78], showing relatively reduced magmatic-hydrothermal fluids at the Mo mineralization stage. The elevated Mn/Fe ratios in the brine assemblages in the quartz phenocrysts (Figure 7E) might represent an oxidized fluid compared to the earlier brines in the UST.

The Cl/Br ratios in the brines were within the range of natural magmatic fluids [79]; however, they were Cl-rich (Figure 7C). Cl/Br ratios in the brines were relatively elevated in magmatic fluids derived from magma with more crustal components than the fluids from porphyry Cu-Mo-Au deposits, possibly due to more change in mixing with a Cl-rich sedimentary rock such as evaporite [43]. A high crustal component was proposed for the formation of arc granitoids southwest of the Korean Peninsula, including the Sannae-Eonyang granite [24,33,34]. The elevated Cl/Br ratios in the brine in the Eonyang UST coincided with fluids from the magma of a high crustal component.

Trace elemental ratios in the brine assemblages suggested that the UST was formed from a single batch of magmatic fluids, whereas temporally later magmatic fluid hosted in the phenocrysts was derived from geochemically new and relatively oxidized magma, possibly formed by the mixing of a new component of magma or country rocks. This contrasting fluid chemistry of the UST sequence and post-UST strongly suggests that a sequence of a UST band was formed by a single pulse of periodically released magmatic fluids (Figure 11).

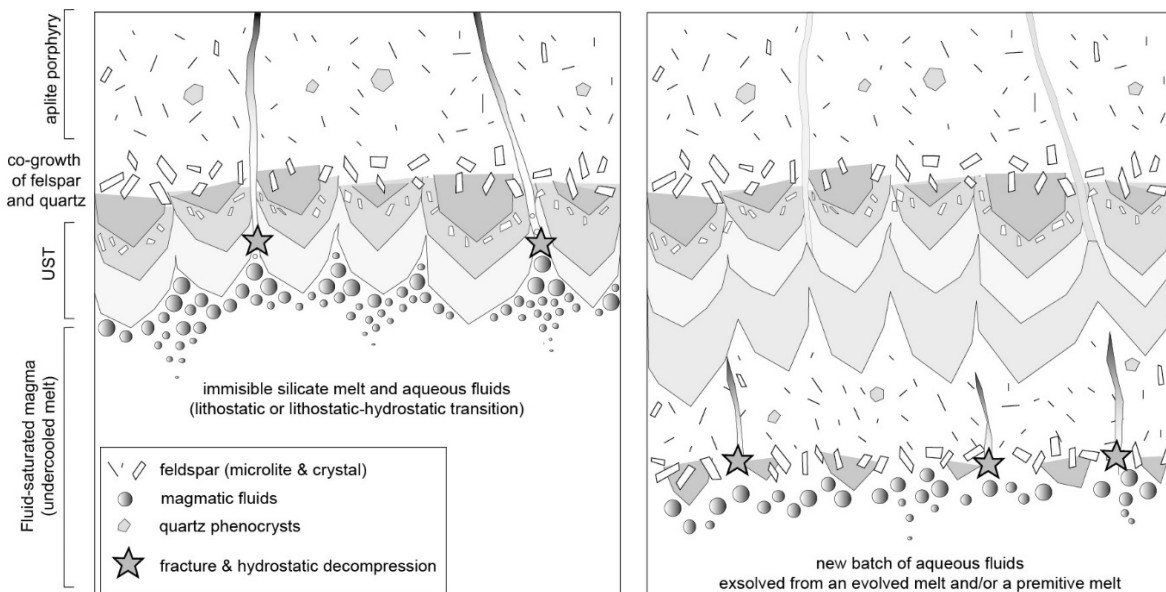

**Figure 11.** Schematic and interpretive drawings of generation of Eonyang UST in the Sannae-Eonyang granite. In the upper part (approximately > 8 km below paleosurface) of the crystallizing granitic magma of the Sannae-Eonyang granite, lithostatic pressure accumulated by aqueous fluid release from an undercooled silicate melt will shift the cotectic boundary from co-growth of feldspar and quartz into quartz-only phase crystallization. Hydrofracturing will cause transient hydrostatic pressures and fluid-phase separation, but the pressure will rise subsequently by the sealing of secondary quartz. Highly fluctuating pressures will continue during the formation of a sequence of UST until the cessation of a single batch of magmatic fluid releasing. After magmatic fluid release, the undercooled and residual melt will be quenched to form a fine-grain aplite porphyry. New batches of magmatic fluid are required to form a later sequence of UST.

### 6.6. Implications for an Economic W-Mo Ore in the Area

Fluid inclusions hosted in quartz veins in the Sannae W-Mo deposit [28] located in the eastern part of the Sannae-Eonyang granite showed a similar phase proportion and distribution of fluid inclusions to the Eonyang UST in the present study. A Boling assemblage has also been reported in quartz veins in the Sannae deposit [27]. Although the Sannae deposit is approximately 12 km away from the Eonyang UST (Figure 1), we speculated that the two hydrothermal systems showed similar magmatic-hydrothermal fluid processes and that it is worthwhile to investigate the mineralization potential in the studied area.

USTs occur in many ore-bearing intrusions. In some magmatic-hydrothermal regions, the UST selectively occurs in ore-bearing intrusions [15,80], suggesting that the UST truly indicates a water-saturated intrusion with a significant mineralization potential. Because the Eonyang UST is formed from a highly fractionated melt, we could expect to see an elevated concentration of incompatible metals such as W and Mo dissolved in the fluid inclusions, including the Sangdong W-Mo deposit in Korea [81,82]. Fluid inclusions in the Eonyang UST provide unambiguous evidence of the exsolution of W- and Mo-rich magmatic fluids. The W and Mo concentrations in the brine assemblages in the UST are 140–410 and 110–590 μg/g, respectively (Table S2). To examine the W-Mo mineralization potential, we examined the geochemistry of W-Mo mineralization. Wolframite, scheelite, and molybdenite precipitation can be promoted by the following chemical reactions:

$$H_2WO_4 \text{ (aq)} + FeCl_2 \text{ (aq)} \rightarrow FeWO_4 + 2HCl \text{ (aq)} \tag{1}$$

$$H_2WO_4 \text{ (aq)} + CaCl_2 \text{ (aq)} \rightarrow CaWO_4 + 2HCl \text{ (aq)} \tag{2}$$

$$H_2WO_4 \text{ (aq)} + CaCO_3 \rightarrow CaWO_4 + CO_2 + H_2O \tag{3}$$

$$H_2MoO_4 \text{ (aq)} + 2H_2S \text{ (aq)} + H_2 \text{ (aq)} \rightarrow MoS_2 + 4H_2O \tag{4}$$

Wolframite and scheelite precipitation from the $W^{6+}$ complexes [83] require dissolved cations such as Fe (or Mn) and Ca (reactions 1–3). High Fe (4.9–20.0 wt.%) and Mn (4.1–12.0 wt.%) concentrations in the brines in the UST (Table S2) and Ca content (CaO of 0.5–4.3 wt.%) in the Sanne-Eonyang granite [26,36,37,84] allow precipitation of the W-Mo minerals if HCl in the fluids can be eliminated (reactions 1–2). The pH of the fluids can be easily controlled by a destructive alteration of feldspar to muscovite [85] which is abundant in the Sannae-Eonyang granite. Molybdenite is precipitated from $Mo^{6+}$ [86–88] dissolved in the fluids and reacts with reduced sulfur species (reaction 4). Although it is difficult to estimate the redox state of the fluids, 0.7–1.8 wt.% of S in the brines we found in the Eonyang UST (Table S2) would be sufficient to precipitate molybdenite. Mn/Fe ratios in the UST brines and phenocrysts are also relatively reduced during the earlier UST stage. In the Eonyang UST, the geochemistry of the UST fluids indicates the possibility of a hidden W-Mo mineralization. While economic W-Mo mineralization does not occur nearby, or may be hidden, a further detailed study is required to investigate the Eonyang area.

## 7. Concluding Remarks

Our microanalytical study of melt and fluid inclusions in the UST in the Eonyang area suggests that periodic fluid exsolution from the upper part of the crystallizing magma created a UST band sequence (Figure 11). At depths > 8 km below the paleosurface, a pulse of magmatic fluids created a transient pressure up to 2.3 kb, which allowed the shifting of cotectic boundaries for quartz crystallization in the UST. Replacement of hydrostatic-lithostatic pressure continued during UST formation until the cessation of magmatic fluid release (Figure 11). Such a highly fluctuating pressure during the UST formation could cause a large range of salinities in the exsolving magmatic fluids. The extremely low temperature range of approximately 320–390 °C recorded in the fluids indicated a strongly undercooled disequilibrium during UST formation. A new batch of magmatic fluids released from possibly a new fluid-saturated magma would be required to create the next

sequence of the Eonyang UST. Metal concentrations in the fluids in the Eonyang UST suggest the potential of hidden W-Mo mineralization in the area.

**Supplementary Materials:** The following are available online at https://www.mdpi.com/article/10.3390/min11080888/s1, Figure S1. Na-normalized ratios of trace elements, including K, Fe, Mn, Cl, Br, Zn, Ba, Pb, Mg, Sn, Rb, Sr, Cd, As, Sb, S, Bi, Tl, In, Ge, Y, La, Ce, U, and Th in brine inclusion assemblages in the UST quartz stages (1–6) and quartz phenocrysts. Figure S2. Harker diagram of major elements and trace element ratios in the Sannae-Eonyang granite [26,36,37]. Mg/Al$_2$O$_3$ and Nb/TiO$_2$ ratios obtained from silicate melt inclusions (SMI) in the quartz phenocrysts led to approximate SiO$_2$ contents (approximately 76–80 wt.%) during Eonyang UST formation. Table S1. Microthermometry results for brine and liquid-rich inclusion assemblages. Apparent salinities (wt.% NaCl equivalent), densities, and pressures were calculated using a model H$_2$O-NaCl system (Bodnar, 1994; Driesner and Heinrich, 2007). The depth was calculated from lithostatic pressure (density of 2.7 g/cm$^3$) and hydrostatic pressure (density of 1 g/cm$^3$). Table S2 Element concentrations (in ppm) in fluid and melt inclusion assemblages using LA-ICP-MS microanalysis. Table S3 LA-ICP-MS analysis of quartz (ppm). An internal standard of stoichiometric Si content in quartz was applied. The red color indicates it is below the detection limit, and its number represents the detection limit.

**Author Contributions:** Conceptualization, J.H.S.; fieldwork and petrography, J.H.S., Y.K. and T.L.; microthermometry, Y.K. and T.L.; Raman, Y.K.; LA-ICP-MS, J.H.S., T.L. and M.G.; original draft preparation, J.H.S.; review and editing, J.H.S. and M.G. All authors have read and agreed to the published version of the manuscript.

**Funding:** This work was supported by a National Research Foundation of Korea (NRF) grant funded by the Korean Government (MSIT) (No. 2019R1C1C1002588 and 2020R1A6A3A13075639).

**Data Availability Statement:** Not applicable.

**Acknowledgments:** We acknowledge Laurent Oscar for assisting with the LA-ICP-MS analyses at ETH Zurich and John H. Dilles (Oregon State University) for his advice on the manuscript. The suggestions and advice from the two reviewers are acknowledged.

**Conflicts of Interest:** The authors declare no conflict of interest.

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
