# Peer review of "Periodically Released Magmatic Fluids Create a Texture of Unidirectional Solidification (UST) in Ore-Forming Granite: A Fluid and Melt Inclusion Study of W-Mo Forming Sannae-Eonyang Granite, Korea"

_minerals, doi:10.3390/min11080888_

Round 1
Reviewer 1 Report
Compared to a previous version, the manuscript quality increased substantially. The issues that could be reasonably solved have been solved. I am satisfied with the response to my initial comments and requests.
I only recommend that the manuscript is looked at one more time to correct for sentence structure and grammar. See some suggestions in the PDF document attached, but there might be others as well.
Good luck with this interesting paper!

Author Response
I revised it as suggested. Thank you very much!

Reviewer 2 Report
This is a paper presenting a fluid and melt inclusion study of W-Mo forming Sannae-Eonyang granite, Korea, which indicating a periodically released magmatic fluids create a texture of unidirectional solidification in ore-forming granite. Combining field observations and petrogenesis, melt/fluids inclusions of quartz and quartz phenocrysts is set out on a new batch of magmatic fluids released from possibly a new fluid-saturated magma. The paper need to be polished. Here, some majior changes should be required, as follows:
1. Periodically Released Magmatic Fluids? It is difficult to understand what is a periodic fluid? What is reasons? Resulted from mixtures of arc magma with continental crust? or from the growth kinetics of quartz?
2. The UST quartz samples mainly come from cavities or near cavities, crystallized relatively later. Therefore, the measured temperature does not represent the emplacement condition of granite, T is very low, so it is inappropriate to discuss the whole process.
3. In Fig. 7, why there is no dashline between the stage 3 and stage 4?
4. The ore deposits and petrological characteristics related to tungsten and molybdenum are not introduced in the front of the article. But in the discussion part, you discuss tungsten and molybdenum mineralization, it is abrupt.
5. From the trace elements of quartz, it was formed relatively later, and is enriched in Li, B, etc. indicating that it was formed in a fluid rich environment. It is very inappropriate to use these data to discuss the periodically released fluid from magma.
Author Response
Point 1: Periodically Released Magmatic Fluids? It is difficult to understand what is a periodic fluid? What is reasons? Resulted from mixtures of arc magma with continental crust? or from the growth kinetics of quartz?.
Response 1: Thank you for the inquiry. What I meant here in the paper about the “periodic fluids” is that a magmatic fluid releasing from an intrusion is not very continuous, but highly episodic. This might be due to a magma mixing, a volatile saturation, or both of the events. Here I studied an intrusion of arc magma mixed significantly with crustal components (Sannae-Eunyang granitoids), but I think the periodic fluid release can be a very general feature in a fluid-releasing magma since the UST textured granitoids are not so uncommon in many types of magmatic-hydrothermal ore deposits globally.
Point 2: The UST quartz samples mainly come from cavities or near cavities, crystallized relatively later. Therefore, the measured temperature does not represent the emplacement condition of granite, T is very low, so it is inappropriate to discuss the whole process.
Response 2: Thank you for your points. Yes the cavities and the UST are the later-stage magmatic process, but I think P-T condition during the UST formation possess some important implications for the late-stage granitic magmatism like the formation of pegmatite and magmatic-fluid releasing. In terms of pressure: the pressure and the calculation of a paleodepth during the formation of the UST are, although it is a late-stage, I think not much different from an earlier stage of the magmatism. In terms of temperature: temperature during the UST formation may be significantly different from the earlier stage of the magmatism. Since UST is an important late-stage magmatic feature similar to a pegmatite, we showed in the manuscript that the significantly lower range of temperature implies a significant undercooling during the latest stage of granitic magmatism, and the temperature range might bridge a (temperature) gap between the granitic magma and the magmatic-hydrothermal ore formation.
Point 3: In Fig. 7, why there is no dashline between the stage 3 and stage 4?.
Response 3: Thank you. I add dashlines now.
Point 4: The ore deposits and petrological characteristics related to tungsten and molybdenum are not introduced in the front of the article. But in the discussion part, you discuss tungsten and molybdenum mineralization, it is abrupt.
Response 4: Thank you, but I introduced it in the second paragraph of the introduction. I slightly revised it to make it clearer.
Point 5: From the trace elements of quartz, it was formed relatively later, and is enriched in Li, B, etc. indicating that it was formed in a fluid rich environment. It is very inappropriate to use these data to discuss the periodically released fluid from magma.
Response 5: Thank you for your point. I add the trace element concentrations in quartz because it indicates a quartz growth rate if it is not attaining an equilibrium. If you see figure 8, you can see abrupt concentration drops of Al and Ti between the end of the UST stage and the phenocryst stage. It might represent an increasing growth rate from quartz stage 1 to stage 6 (UST) dropping at the phenocryst stage, which might also be something to do with a periodic fluid-releasing.

Round 2
Reviewer 2 Report
This version has been greatly improved, but the authors can not explain my question. In view of the characteristics of the article, I suggest publishing after minor revision. Such as: page 21, the last [1] for 1. [2] for [2].This manuscript is a resubmission of an earlier submission. The following is a list of the peer review reports and author responses from that submission.
Round 1
Reviewer 1 Report
See attached PDF file with numerous edits and comments. Here I insert detailed recommendations for improvement.
Overall, this inclusion study would be an important contribution that should interest a broad community of high-temperature geochemists/inclusionists, ore geologists, and petrologists. The study focuses on the fluid evolution of quartz-feldspar granitic assemblages using combined fluid and melt inclusions analysis. In contrast to hydrothermal veins and texturally homogeneus granitoids, rocks that form at the magmatic hydrothermal transition are relatively poorly studied, and for this reason alone, the study is significant and necessary. However, I recommend publication, but only after some important clarifications and numerous suggested edits are made.
Major issues:
- It is concerning that the study is only based on one sample, apparently and it does not include any details or photographs from the field.
- The “sample” should be put in context of its location in the field.
- What is the size of the sample?
- How common is this UST quartz and how is it associated with the mineralization? Based on figure 1, it seems like the UST quartz is located >20 km away from the W-Mo deposit, so it’s not really associated with it as alluded in the introduction
- More importantly, how far is this sample from the contact of the intrusion with country rock; Was there a chance that the magma experienced very rapid cooling?
- How wide are the pegmatite-aplite, UST, and porphyritic intervals? Give numerical values. Characterize these bands in the field. For how long can they be traced in the outcrop? Do they occur near the contacts of the intrusion, footwall or hanging-wall, or are they pervasive, through the entire intrusion?
- What is the “pegmatite-aplite interface”? What is considered to be “pegmatite” in this paper? Only the quartz crystals shown in Fig. 2, or are there other minerals exceeding 2.5 cm, like feldspar, mica, etc.?
- Some overview field photos would be important to clarify and complement the field description of these textural varieties.
- The occurrence of silicate melt inclusions is insufficiently characterized and not convincing.
- What are the ratios between fluid and solids in the crystallized SMIs?
- Provide higher resolution photos of increased magnification with individual phases labeled. Fig. 3F, for example, does not allow to see anything in the dark portion of the melt inclusion.
- Include some Raman ID of mineral phases to characterize the melt inclusions and to demonstrate that they are trapped granitic melt and not merely microcrystals that were accidentally trapped along with fluid.
- Is there a host-correction for SiO2 concentration in the MI laser ablation data? Only three inclusions were analyzed by LA-ICP-MS and only from two quartz phenocrysts. Melt inclusions from the UST quartz were not analyzed. How representative were these inclusions?
- Terminology corrections or clarifications are needed and should be corrected throughout the paper:
- Using “aqueous liquid inclusions” and “vapor inclusions” is ambiguous, even though the expressions are explained on page 5. All the “liquid inclusions” contain a vapor bubble and the “vapor inclusions” contain a liquid phase at ambient conditions. I would request to substitute these everywhere with “liquid-rich inclusions” and “vapor-rich inclusions”.
- Inclusion petrography and microthermometric results
- How are the “pseudo-secondary” trails defined? Are the fractures limited to one quartz type? How long are they? Based on the evidence presented how can one distinguish them from secondary trails?
- It is unclear to me how the FIAs are defined and separated. In section 3.2, it is stated “based on the texture of quartz zones and the trails of the FIAs”. But in Table S1, the same FIA# repeats multiple times for different quartz zones. For example FIA# “brine 1” shows up in both stage 1 and in quartz phenochryst 1 and 2. Does this mean that the same fracture trail cuts through all these different quartz types? Wouldn’t that be a secondary trail rather than a pseudosecondary trail?
- In Table S1 include a column with number of inclusions analyzed for each FIA.
- It appears that no vapor inclusions have been measured (bottom of page 6 and Table S1). In my opinion this is an important deficiency of the study. Their homogenization temperatures should be similar to the co-existing brine inclusions in order to properly document fluid boiling (immiscibility).
- Although the paper presents results for FIAs, in Fig. 10 it appears that isochores are plotted for each fluid inclusion. Ideally, the isochores for the boiling assemblages should start at the T and P of homogenization and up (only above the solidus). The corresponding vapor-rich inclusions within the boiling FIAs should start at about the same T and P as the brine inclusions, if indeed the system experienced fluid immiscibility, and they were not caused by some post-entrapment modification. But unfortunately the vapor-rich inclusions were not measured or data were not included.
- Explain what equations of state were used to calculate the isochores and why the vapor-rich inclusions are missing.
- Big picture interpretations. An important point that results from this investigation is that, if melt inclusions are indeed present in both the UST quartz and phenochrysts, this system demonstrates a strongly undercooled magma. The authors should include this very important conclusion in their study: According to Fig. 10, the actual temperatures at which crystallizing magma still existed would be far below the equilibrium of the water granitic solidus, ranging ~350 to 450°C based on the boiling assemblage, which is stated not to need a pressure correction.
- These temperatures are extremely low and they support undercooled disequilibrium crystallization of the magmatic system. The samples studied here include granophyric texture (a type of skeletal texture) and the UST quartz which also seems to be skeletal in nature. These are also considered to be evidence of strongly undercooled disequilibrium. See London’s numerous papers on pegmatites, Sirbescu et al. (2017), Journal of Petrology, Nabelek et al. (2010) Contributions to Mineralogy and Petrology, Ackerson et al. (2018) Nature, and Rusiecka and Baker (2021) Contributions to Mineralogy and Petrology as a few examples.
- The consequences of these very low temperatures would be that the diffusion rates through the chilled magma would be very low. Even migration of exsolved bubbles would be practically impossible, because the viscosity of the silicate liquid would be too high. Therefore, in my view, the fluid compositional variations would take place probably only locally, at the scale of centimeters, rather than the whole intrusion. It would be difficult to interpret any causative relation to Mo-W deposit.
Finally, I would request to strengthen this study and address the points above, but in particular the documentation of this occurrence in the field (point 1), and the coexistance of melt inclusions (point 2). If there is enough evidence to show that very rapid cooling is possible, this would be an important additional evidence that felsic magmas can experience very low temperatures, hundreds of degrees below what is considered equilibrium minimum temperature of crystallization (point 5).

Reviewer 2 Report
I originally prepared two separate documents (annotated PDF for minor comments and a more extensive document with general comments).
Since I apparently cannot upload more than one additional file as a reviewer, I have combined these two PDFs into a single file. Please refer to this for my comments. My apologies for the inconvenience.
